# Updates on the Management of Colorectal Cancer in Older Adults

**DOI:** 10.3390/cancers16101820

**Published:** 2024-05-10

**Authors:** Conor D. J. O’Donnell, Joleen Hubbard, Zhaohui Jin

**Affiliations:** 1Mayo Clinic School of Graduate Education, Mayo Clinic College of Medicine and Science, Mayo Building, Rochester, MN 55905, USA; odonnell.conor@mayo.edu; 2Allina Health Cancer Institute, Minneapolis, MN 55407, USA; 3Division of Medical Oncology, Mayo Clinic, Rochester, MN 55905, USA

**Keywords:** colorectal cancer, geriatric oncology, comprehensive geriatric assessment

## Abstract

**Simple Summary:**

The majority of cases of colorectal cancer occur in those over the age of 65. Treatment of colorectal cancer in older adults warrants specific considerations due to the effects of aging on comorbidities, functional and cognitive status, and socioeconomic factors. Several recent advances have been made to improve oncological outcomes and reduce toxicities with colorectal cancer treatments in both localized and metastatic disease settings. This review highlights the importance of comprehensive geriatric assessment and provides recommendations to guide the management of older adults with colorectal cancer. It summarizes prospective data from recently reported clinical trials focused on older adults. Other recommendations must at times rely on extrapolations and post hoc analyses due to the underrepresentation of older adults in colorectal cancer trials. This review should also, therefore, serve as a call to action for the field to increase the representation of this substantial and often vulnerable group of patients in future colorectal cancer trials.

**Abstract:**

Colorectal cancer (CRC) poses a significant global health challenge. Notably, the risk of CRC escalates with age, with the majority of cases occurring in those over the age of 65. Despite recent progress in tailoring treatments for early and advanced CRC, there is a lack of prospective data to guide the management of older patients, who are frequently underrepresented in clinical trials. This article reviews the contemporary landscape of managing older individuals with CRC, highlighting recent advancements and persisting challenges. The role of comprehensive geriatric assessment is explored. Opportunities for treatment escalation/de-escalation, with consideration of the older adult’s fitness level. are reviewed in the neoadjuvant, surgical, adjuvant, and metastatic settings of colon and rectal cancers. Immunotherapy is shown to be an effective treatment option in older adults who have CRC with microsatellite instability. Promising new technologies such as circulating tumor DNA and recent phase III trials adding later-line systemic therapy options are discussed. Clinical recommendations based on the data available are summarized. We conclude that deliberate efforts to include older individuals in future colorectal cancer trials are essential to better guide the management of these patients in this rapidly evolving field.

## 1. Introduction

Colorectal cancer (CRC) is the second most deadly and third most commonly diagnosed cancer worldwide [1]. The risk of CRC increases with each decade of life. More than two thirds of CRC cases occur among those 65 years of age and older [2].

Important strides have been made in recent years to increase efficacy or reduce toxicity in the treatment of both early and advanced CRC. However, there often remains limited prospective data to guide the management of older patients with CRC [3]. Older patients are frequently underrepresented in or excluded from clinical trials. Under these circumstances, treating clinicians must often rely on the interpretation of pooled analyses. As the discipline of geriatric oncology has grown, there has been some emergence of prospective trials focusing specifically on older adults; however, these often include patients with a wide range of malignancies. Those trials looking specifically at interventions for older patients with CRC are few in number and often of smaller sample sizes.

When caring for the aging patient, additional complexities, including age-related comorbidities, cognitive and functional impairments, and socioeconomic constraints, may need to be considered. This article will highlight current advances and challenges in the management of older individuals with CRC. Specifically, we will examine the evidence that modern treatment approaches effectively improve quality of life and longevity in elderly patients with CRC. A focus will also be made on highlighting strategies of treatment de-escalation that may be appropriately considered to minimize toxicity in older individuals who are less fit.

## 2. Geriatric Assessment

In geriatric oncology, functional rather than chronological age should determine management, as chronological age alone is a poor predictor of cancer treatment tolerance [4]. A great deal of effort should be dedicated to evaluating functionality and maintaining it during the treatment of older adults. When the body is put under stress from disease or its treatments, functional limitations may be unmasked. This process of functional decline is expressed clinically as frailty, a state of increased vulnerability to stress with an increased risk of adverse outcomes during cancer treatment [5].

Various frailty scores have been developed, and they clearly correlate with poorer overall survival [6,7]. These screening tools (Table 1), such as the G8 [8,9], may be used to identify patients in need of a more comprehensive geriatric assessment (CGA). 

A CGA is a multidisciplinary diagnostic process focused on determining a frail older person’s medical, psychological, and functional capacity, with the goal of developing a coordinated and integrated plan for treatment and long-term follow up. The various assessment domains and common potential interventions are outlined (Table 2). CGA use is recommended in various guidelines [14,15]. Although the CGA calls for an individualized approach to patient care, for the purposes of discussion and recommendation of CRC therapies, we may consider patients on a spectrum: functionally independent (fit), those with vulnerability/frailty (medium-fit), and finally those with severe limitations, with no possibility of recovering functional reserves and a limited life expectancy (unfit).

A growing body of evidence shows CGA effects on oncological and non-oncological treatment decisions (Table 3) [22]. CGA is associated with improved communication and advanced care planning with patients [21,23,24]. CRASH [25] and CARG scores [26], which use various parameters determined during a CGA, may predict severe toxicity in older patients with cancer treated with systemic therapy. They have similar predictive performance [27].

The GAIN trial was one of the first randomized trials to show the benefit of geriatric intervention prior to the initiation of chemotherapy for solid tumors in older adults [21]. Grade 3–5 adverse events were decreased by 10% (60.6% in the standard of care arm compared to 50.5% in the intervention arm, *p* = 0.02) by the addition of geriatric intervention. The finding that initial chemotherapy doses, rates of dose modification, and discontinuation were not different between the two groups highlights that toxicities can be reduced through implementing geriatric interventions other than decreasing chemotherapy. The rates of emergency department visits, unplanned hospitalizations, and survival did not differ during follow-up.

The GAP 70+ study similarly showed geriatric interventions can diminish the rate of grade 3–5 adverse events in elderly patients with advanced disease and at least one impaired geriatric assessment domain [28]. Grade 3–5 adverse events were reduced by 20% (71% in standard of care vs. 51% in intervention arm). Unlike the GAIN trial, treatment intensity was reduced at cycle one (i.e., primary dose reduction) in the geriatric intervention group more commonly. Importantly, reduced dose intensity in the intervention group did not compromise survival, which was similar between the study groups. Patients in the CGA group also had fewer falls and more medications discontinued, reducing polypharmacy. 

The INTEGERATE study is another randomized trial investigating CGA in cancer patients [29]. They were able to show a benefit in quality-of-life scores, including better scores in functioning, mobility, burden of illness, and future worries from integrating CGA into oncology care. A reduction in emergency department visits and hospital admissions was seen in this study. These results are in contrast to those of the Canadian 5C trial, where no quality-of-life benefit was identified from geriatric assessment and management intervention [30]. The lack of benefit in the 5C study could be because the geriatric assessment was not done until the day of or after first systemic therapy, and only 1.7% of patients had their initial cancer treatment plan modified by geriatric assessment. 

CGA has also been studied in oncological surgery settings. A recently published trial randomized patients ≥ 65 years of age planning for gastrointestinal cancer surgeries to perioperative geriatric consultation or usual standard of care [31]. The trial failed to meet its primary endpoint of reducing post-operative length of stay with the intention of treating the population. However, only 43% of patients in the experimental arm attended their pre-operative assessment. Geriatric intervention led to a significant reduction in post-operative length of stay (5.90 vs. 8.21 days, *p* = 0.024) and a trend towards lower rates of postoperative intensive care use, 90-day readmissions, and major complications in the intervention group in the per protocol analysis. Similar to the 5C trial, these findings highlight the need for coordination of appointments when assessing geriatric interventions in future clinical trials. Consideration of telehealth and other measures to reduce the burden of additional clinical visits for older cancer patients should be explored.

Other retrospective data has suggested decreased 90-day mortality from geriatric peri-operative co-management with surgeons in those undergoing surgery for cancer (odds ratio [OR], 0.43 [95% CI, 0.28–0.67]; *p* < 0.001) [32].

Despite the emerging data showing the benefits of geriatric assessment in the management of older adults with cancer, uptake in clinical practice often remains somewhat limited. Several challenges persist in relation to the time and resources needed to complete a CGA, which takes approximately one hour initially and requires follow-up. There is a general and increasing shortage of geriatricians in the United States [33]. However, the implementation of comprehensive geriatric oncology care may provide a higher-value (quality/cost) intervention compared to many more recently approved pharmaceutical interventions [34]. 

Examples of CRC-specific trials exploring geriatric interventions are highlighted within the text.

## 3. Localized Rectal Cancer with Microsatellite Stability

In recent years, evidence for several new approaches to manage stage I (T1–T2 without nodal involvement) and stage II–III (T3–T4, or any T with nodal involvement) rectal cancer has emerged. The inclusion of and sequencing of different treatment modalities (chemotherapy, radiation, and surgery) in management plans is often spiritedly debated at multidisciplinary tumor boards. Older, more frail patients can certainly add yet another layer of complexity to these decisions. 

### 3.1. Early Stage

For early stage rectal tumors, more minimally invasive procedures to avoid the morbidity associated with total mesorectal excision (TME) surgery are being explored. Current National Comprehensive Cancer Network (NCCN) guidelines suggest that transanal local excision may be appropriate for certain mobile T1 tumors [13]. If these criteria cannot be met, generally more extensive surgery with TME with or without radiation has been considered the standard of care. TME involves major surgery with either a lower anterior resection (LAR) or an abdominal perineal resection (APR), the latter of which always requires a permanent end colostomy. A systematic review including 36,315 patients found age > 65 was associated with a statistically significant increase in both anastomotic leak and post-operative death [35].

The GOSAFE study was a prospective observational study of cancer surgery in older patients (age ≥ 70) that assessed patient-reported outcomes post-operatively [36]. They were able to show that major colon and rectal surgery in older patients with CRC results in restoration of good quality of life following surgery in the majority of cases [37]. Furthermore, functional recovery to remain independent occurred in 78.6% (254/323) of patients with colon cancer and 70.6% (94/133) with rectal cancer post-operatively. The authors did note that an Eastern Collaborative Oncology Group (EGOG) score ≥ 2 preoperatively in patients with rectal cancer strongly predicted a decline in quality of life post-operatively (OR 3.81, *p* = 0.006). These results should help inform shared decision-making between patients and surgeons.

Surgery will have a role in many cases of early and locally advanced rectal disease, but the safe expansion of minimally invasive procedures in early disease is an appealing prospect, especially for older individuals with poor performance status.

Some phase II trials have assessed the feasibility of local excisions in T1-3 tumors. The TREC trial randomized 55 patients with cT1-T2, N0 rectal tumors to short-course radiation followed by transanal endoscopic microsurgery after 8–10 weeks (with TME reserved for those with high-risk histopathological features upon excision) versus upfront TME [38]. The study also included a cohort of 61 non-randomized patients considered to be high risk for conventional TME. These patients were older (median 74, oldest 89) and were more likely to have life-limiting comorbidities. A total of 70% of the experimental group and 92% of the non-randomized group were able to avoid TME, and a complete pathological response was achieved in 26% and 31%, respectively, with short-course radiation. An additional 4% of patients in the experimental group and 8% of patients in the non-randomized group achieved a complete clinical response without undergoing transanal endoscopic microsurgery. In the predominantly older, non-randomized cohort, local recurrence-free survival was excellent, with a rate of 91% at 3 years. Extensive quality of life data suggested that organ-preserving therapy was well tolerated, with slight worsening of fatigue, physical, social, role, and bowel functions at three months, but with values largely returning to pre-treatment baseline by 6–12 months.

The CARTS study was a similar single-arm trial where cT1-T3, N0 rectal cancer patients were treated with long-course chemoradiation (CRT) followed by transanal endoscopic microsurgery in the case of a good response. The majority of patients were able to avoid major surgery, with recurrence rates similar to those historically seen with TME. However, a major burden of low anterior resection syndrome (LARS), characterized by the clustering of stools, soiling, and fecal incontinence, was present in 50% of patients at two years. 

In the phase II NEO study, cT1-3abN0 low- or mid-rectal tumors were treated with 3 months of chemotherapy followed by transanal endoscopic surgery [39]. A total of 57% achieved a favorable response to chemotherapy, allowing for organ preservation per protocol. At one year, the rate of major LARS was 14%, with minimal changes in quality of life.

On the whole, this data suggests there may be a role for more limited local excisions to treat CRC in older patients with poor performance status and early stage rectal cancers. This does not guarantee less morbidity, as is particularly seen in those treated with CRT.

### 3.2. Locally Advanced

For more locally advanced rectal cancers, until recently, the standard of care was neoadjuvant radiation (short or long-course CRT), followed by TME surgery, followed by adjuvant chemotherapy for an additional 4 months. In a retrospective Canadian study, patients 70 years of age or older with locally advanced rectal cancer seemed to have similar disease-free survival (DFS) and overall survival (OS) outcomes compared to those younger than 70 [40]. This was the case despite older patients being less likely to complete neoadjuvant CRT and less likely to receive adjuvant chemotherapy.

The PRODIGE 42/GERICO 12 study provides some prospective randomized phase III data specific to individuals ≥ 75 years of age with resectable rectal tumors that were T3-T4 (with or without nodal involvement) or very low T2 tumors [41]. Patients were randomized to neoadjuvant short-course radiation or CRT, followed by surgery. The primary endpoints were R0 resection rate (non-inferiority design) and maintenance of autonomy in instrumental activities of daily living (IADLS) scores. The study only enrolled 105 of the planned 420 participants. A total of 95.6% of patients in the CRT arm versus 88.0% of patients in the SCRT arm had R0 resections, failing to meet non-inferiority for SCRT. Overall, there was a significant difference in the degradation rate of IADLS at month 3 (deterioration in 44% of CRT versus 14.8% in SCRT); however, this difference did not persist at 6 months. Secondary outcomes of OS (HR 0.28, *p* = 0.05) and cancer-specific survival (HR 0.21, *p* = 0.027) appeared to be improved with short-course radiation. We cannot conclude short-course radiation is superior to long-course CRT in older patients based on this limited data, although a larger trial to properly assess OS is warranted given these intriguing results.

Guidelines are, however, shifting to reflect a preference towards total neoadjuvant treatment (TNT) approaches for most patients with locally advanced rectal cancer [13]. TNT has several potential benefits, including higher rates of treatment completion, a reduction in rates of distant metastases, and emerging data to suggest improvements in overall survival for at least one TNT regimen [42,43,44,45,46]. Conversely, there is simultaneously a push for novel approaches focused on omitting some potentially morbid treatment components without compromising outcomes. This may be especially prudent for older individuals, and this population deserves specific attention in future trials.

Earlier trials showed neoadjuvant radiation improved local recurrence rates in locally advanced rectal cancer from 25–40% down to 5–9% [47,48]. However, the treatments are associated with grade 3–4 acute (any 27%, diarrhea 12%) and long-term (any 14%, strictures 4%, bladder problems 2%, gastrointestinal effects 9%) toxicities [49]. In the modern era of magnetic resonance imaging (MRI) staging and TME surgery, radiation may not be necessary for all locally advanced patients. The recently published PROSPECT trial asked the question of whether radiation could be avoided in carefully selected patients in favor of perioperative chemotherapy and surgery alone [50]. Patients with cT2N1 or T3N0-1 without compromised mesorectal fascia eligible for sphincter-sparing surgery were enrolled. The median age was 57, with the oldest enrolled patient being 91 years of age. In the experimental arm, those that responded to neoadjuvant fluoropyrimidine and oxaliplatin (FOLFOX) chemotherapy could proceed to surgery without radiation. Non-inferiority of disease-free survival (80.8% at 5 years) compared to the traditional CRT approach was reached. The 5-year OS rate of 90% was similar between arms, and less than 2% had local recurrence. Importantly, at 12 months, patient-reported outcomes suggested significantly lower rates of fatigue, neuropathy, better bowel function, and better sexual function in the chemotherapy-only group. Although this represents a good potential option for fitter older adults to avoid radiation complications, the full 12 cycles of FOLFOX recommended in this trial may still represent overtreatment for some patients.

In the OCUM study, cT2 tumors, or cT3 tumors of the mid-high rectum were treated with upfront surgery if there was >1 mm distance between the tumor, tumor deposits, or suspicious lymph nodes and the mesorectal fascia [51]. Only 35.7% of these patients went on to receive adjuvant therapy. The local recurrence rate of only 2.9% and the distant metastasis rate of 15.9% at 5 years show excellent outcomes are possible with upfront surgery and limited chemotherapy/radiation use in selective patients. We do not have a comparison of study results based on age for the PROSPECT or OCUM trials [50,51].

Observational cohorts provided some evidence that those with a complete clinical response to neo-adjuvant treatments had promising results despite the omission of upfront surgery. The incidence of regrowth at 2 years was 25.2%, with a rate of distant metastases of 8% and disease-specific survival of 94% in one such cohort of 1009 patients [50]. These findings prompted the Organ Preservation for Rectal Adenocarcinoma (OPRA) study, which randomized patients with low-lying stage II and III rectal cancer to chemotherapy followed by CRT or CRT followed by chemotherapy [46]. The results suggest that 53% of patients treated with CRT followed by 4 months of oxaliplatin-based chemotherapy may be able to avoid surgery without compromising DFS compared to historical norms. The recently published OPERA study looked to further improve the rates of organ preservation for T2-3bN0 (or N1 node < 8 mm) low-mid rectal tumors by incorporating radiation dose escalation [52]. Reported 3-year organ preservation rates achieved by combining standard chemoradiation with local brachytherapy or external beam boost were 81% and 59%, respectively. Although these approaches seem enticing for older people with low rectal tumors, it should be noted that the oldest patients enrolled in these trials were 68 (OPRA) and 79 (OPERA). Furthermore, the social context of each patient should be considered, as close monitoring and frequent assessment following TNT are essential for at least the first 2–3 years after therapy to identify regrowth requiring surgical intervention.

The PRODIGE 23 trial has shown the benefit of neoadjuvant triplet chemotherapy in a highly selective patient population [43,45]. Patients > 75 years of age or with a history of ischemic coronary disease or grade 2 or worse neuropathy were excluded. The trial randomized patients to 6 cycles of fluoropyrimidine plus oxaliplatin plus irinotecan (FOLFIRINOX), followed by long-course CRT, followed by surgery. The protocol also included adjuvant treatment. Compared to the standard of care arm (CRT, surgery, adjuvant chemotherapy), the pathological complete response rate was more than doubled (27.8% vs. 12.1%). A recent update shows a 7-year metastasis-free survival of 73.6% vs. 65.4% and an overall survival of 81.9% vs. 76.1% in favor of the TNT arm. Grade 3–5 adverse events were 47% in the FOLFIRINOX arm [45]. This regimen can be considered for older individuals who meet the inclusion criteria described.

Finally, the RAPIDO trial compared an approach involving short-course radiation followed by neoadjuvant chemotherapy (FOLFOX for 9 cycles or capecitabine plus oxaliplatin (CAPOX) for 6 cycles) only in those patients with locally advanced rectal cancer with high-risk features [42]. Around 40% of patients were over 65 years of age in this trial. At 5 years, disease-related treatment failure was 27.8% (HR 0.79 compared to the standard of care arm). Distant metastases at 3 years were improved by about 8% (HR 0.69) [44]. These findings were achieved without affecting quality of life. However, adoption in some countries has been slow given the concern for an increased rate of locoregional failure (11.7% vs. 8.1% at 5 years).

Although the results of these recent total neoadjuvant approaches are promising, an individualized discussion is warranted. There is potential for overtreatment of certain patients. Clear expectations for potential benefits and harms must be laid out. It bears mentioning that modern adjuvant trials in rectal cancer have largely failed to accrue and have not yielded consistent benefits from systemic therapy in terms of DFS and OS [53]. The ADORE trial is one example where the potential benefit was seen for adjuvant FOLFOX compared to a bolus fluoropyrimidine regimen in those with residual disease after CRT [54]. The DFS advantage appeared to hold up in an exploratory subgroup analysis of patients aged > 65. In general, neoadjuvant is favored over adjuvant chemotherapy on the basis of the studies mentioned above.

## 4. Localized Colon Cancer with Microsatellite Stability

Adjuvant chemotherapy is considered the standard of care for resected stage III colon cancer. Older individuals are less likely to receive adjuvant treatment [55]. However, databases suggest older individuals with CRC who do receive adjuvant therapy do have a survival benefit, with longer treatment duration being associated with decreased mortality [56,57].

Sargent et al. performed a pooled analysis of 7 phase III randomized controlled trials evaluating 5-fluorouracil-based chemotherapy in patients with stage II and III colon cancer [58]. The results demonstrated a clear DFS benefit across all age groups. There was similarly no interaction between age and treatment effect for OS. There was some slight convergence of survival curves at 5 years in the group of patients older than 70 years of age, probably owing to deaths from other causes. Tolerance and toxicity were similar for older and younger adults.

Single Capecitabine and 5-fluorouracil (5-FU) are generally thought to have comparable efficacy, including in older adults [59,60]. The choice between the two can be individualized. 5-FU requires a central line (port-a-cath or peripherally inserted central catheter) for a pump infusion over 46 h. This may present challenges to ambulatory older patients. Capecitabine is administered orally. Although oral administration presents some advantages, oral drugs may be subject to higher copayments depending on the health plan coverage and may be more prone to compliance issues and medication dosing errors. Given its renal excretion, capecitabine may be contra-indicated or require dose adjustment in those with chronic kidney disease. In the MRC FOCUS2 trial of older and frailer adults with metastatic colorectal cancer deemed unfit for upfront full-dose therapy, grade 3–4 toxicities were higher with capecitabine than 5-fluorouracil [61]. Quality of life scores did not differ significantly between the two agents. Capecitabine was associated with higher rates of diarrhea, nausea, and hand-foot syndrome, while 5-fluorouracil caused more stomatitis. Dihydropyrimidine dehydrogenase (DPD) testing can be considered prior to the initiation of either agent [14]. Deficiency is estimated in 8% of Caucasians, warranting dose adjustment. The potential for life-threatening toxicities to 5-fluorouracil or capecitabine may result from treatment in those with severe DPD deficiency.

FOLFOX became a new standard of care for stage III colon cancer in 2004 [62]. At the 10-year follow-up of the MOSAIC trial, the absolute decrease in the risk of death from stage III colon cancer was 8.1% (67.1% alive with FOLFOX versus 59.0% with 5-FU). Absolute recurrence-free survival was also improved by 8.4% (62.2% with FOLFOX recurrence-free versus 53.8% with 5-FU) [63]. The data regarding the use of oxaliplatin in older patients remains mixed, however.

Subgroup analyses within this MOSAIC trial did not show a significant DFS benefit with the addition of oxaliplatin for patients > 70 [62]. The NSABP C-07 trial, which investigated a FLOX regimen (not commonly used in practice nowadays), similarly did not demonstrate a benefit for those over 70 years of age [64,65]. The XELOXA trial comparing 5-FU to a combination of CAPOX showed a trend toward benefit from the addition of oxaliplatin in terms of DFS and OS in elderly patients, although this was diminished compared to younger patients and was not statistically significant [66,67]. 

An earlier publication from the ACCENT database pooled these trials and failed to find evidence of improved DFS and had a trend towards detrimental effects for OS with the addition of oxaliplatin to adjuvant stage II/III colon cancer treatment in patients over 70 years of age [68]. The most recent meta-analysis to address this question comprised five publications, including eight studies with a total of close to 2000 patients [69]. They did not find a statistically significant improvement in DFS or OS by the addition of oxaliplatin to adjuvant therapy in those ≥ 70 years of age with resected high-risk CRC.

A more recent pooled analysis of 12 trials from the ACCENT/IDEA databases did not investigate directly the additional benefit of oxaliplatin in stage III colon cancer but rather compared toxicity patterns, treatment adherence, and outcomes between patients ≥ 70 or <70 years of age receiving oxaliplatin-based regimens [70]. The rates of grade ≥ 3 toxicities were similar with FOLFOX between the two groups; however, CAPOX led to more diarrhea, mucositis, and neutropenia and less neuropathy in older compared to younger patients. There were higher rates of early treatment discontinuation among the older patients (22.0% vs. 15.5%). While traditional outcomes such as DFS and OS were shorter for older patients, the authors highlighted that time to recurrence (TTR) was not statistically different based on age group in multivariable analysis. TTR allows for the direct impact of adjuvant chemotherapy on the recurrence rate to be assessed, by excluding background noise from non-cancer-related deaths. The authors concluded that in older patients fit for clinical trials, oxaliplatin seems safe and effective to reduce recurrence rates.

The ongoing phase III ADAGE trial (ClinicalTrials.gov identifier: NCT02355379) specifically involves patients ≥ 70 years of age with resected stage III colon cancer and may provide the prospective data needed to settle the debate about adjuvant oxaliplatin in this setting.

“Real-world” experience is captured in the Surveillance, Epidemiology, and End Results–Medicare registry. A total of 814 patients older than 65 years of age who received adjuvant oxaliplatin-based therapy for stage III colon cancer were compared to 3581 patients who received fluoropyrimidine-only therapy [57]. Oxaliplatin-based therapy was associated with improved OS (HR 0.73, *p* < 0.001) and CRC-specific survival (HR 0.39, *p* < 0.001) when compared with 5-FU alone. The benefit did not extend to those 80 years of age and older. These results, which suggest older adults may benefit from oxaliplatin, should be interpreted with some caution, as there is likely a selection bias in non-randomized data where fitter patients may be more likely to receive combination therapy. Other studies have shown only about 14% of stage III colon cancer patients 75 years of age or older receive oxaliplatin-based adjuvant therapy [55].

A major step forward to potentially limit the toxicity of oxaliplatin-based adjuvant therapy for stage III colon cancer came with the publication of the IDEA trial [71]. A preplanned prospective pooling of six trials compared the outcomes of 3 versus 6 months of CAPOX/FOLFOX [72]. For the OS analysis, there was no difference between 3 versus 6 months for those ≥ 70 years of age (HR 1.00, 0.88–1.14). Grade 2 or higher neuropathy rates were reduced to roughly 15% from 45% by treating for 3 months versus 6 months. Persistent neuropathy at longer follow-up is also decreased [73]. Guideline recommendations are largely based on an exploratory subgroup analysis of the IDEA trial. For patients with “low risk” disease, 3 months of CAPOX are statistically non-inferior to 6 months. For those with “high risk disease” (classified by T4 and/or N2), non-inferiority was not achieved statistically, but the absolute difference in OS at 5 years was 1% between 3 and 6 months of CAPOX and 2.8% between 3 and 6 months of FOLFOX [74,75]. Other data suggests that oxaliplatin can be dropped after 3 months while continuing the 5-FU/capecitabine component without compromising outcomes [76,77]. Based on the above data, particularly for older patients, if adjuvant oxaliplatin is considered, the potential diminishing returns of therapy beyond 3 months should be discussed. 

Prospective data does exist incorporating the CGA into adjuvant treatment decisions, specifically for patients with high-risk stage II and III CRC. A single institution study enrolled 195 patients ≥ 75 years of age and classified patients as “fit”, “medium-fit”, or “unfit” [78]. In multivariable analysis, they found that this geriatric classification was the only independent factor predictive of overall survival. The “unfit” population was nearly twice as likely to die from non-cancer-related causes compared to cancer-related causes, providing support for the therapeutic decision to not treat with adjuvant therapy in this population.

### Stage II Disease

There is less data to support an OS benefit from adjuvant chemotherapy for stage II colon cancer [79,80]. Generally, for those with high-risk clinical/histopathological features, chemotherapy is considered, but observation is also considered an acceptable option [13]. Not all high-risk features are considered equal, and it should be noted that T4 disease (stage IIB) actually has a poorer prognosis than stage IIIA disease on SEER data, although increased risk does not necessarily translate to increased benefit from adjuvant therapy. The addition of oxaliplatin to fluorouracil did not show a statistically significant DFS or OS benefit for high-risk stage II disease in a post hoc exploratory analysis of the MOSAIC trial [62,63]. This may appropriately lead to further hesitation to prescribe oxaliplatin to older patients with high-risk stage II disease. 

The emerging technology of circulating tumor DNA (ctDNA) has the potential to aid decision-making in this space. There is potential to spare both younger and older patients from chemotherapy without significantly impacting their chance of disease recurrence. ctDNA may be the most significant prognostic factor associated with recurrence for stage II and III CRC [81]. 

In the phase II DYNAMIC study, patients with stage II colon cancer (patients over 70 years of age comprised 27% of the study population) were randomized to either ctDNA-guided management or standard adjuvant management [82]. The ctDNA-guided management group received less chemotherapy than the standard arm (15% versus 28%; RR, 1.82), with non-inferior two-year DFS (93.5% versus 92.4%, respectively). 

In older patients potentially at higher risk of morbidity from adjuvant chemotherapy for high-risk stage II disease, it is reasonable to consider ctDNA testing, if available, to help support a decision to omit therapy. It must be noted, however, that early knowledge of cancer recurrence, as predicted by ctDNA positivity, in the absence of clear, effective interventions has the potential to cause significant distress to patients. The use of these tests is currently not yet endorsed by NCCN or ESMO [13,83] guidelines and requires a nuanced discussion with patients before ordering. 

## 5. Localized Colorectal Cancers with Microsatellite Instability

Arguably, one of the most exciting areas of clinical research for CRC in recent years involves the management of patients with deficient mismatch repair (dMMR) or microsatellite instability (MSI-high) CRC. Late-onset colorectal cancer (occurring in those 80 years of age or older) compared to earlier onset is more likely to be right-sided (82% vs. 35%) and dMMR (35% vs. 8%)—particularly driven by the BRAF V600E mutation (35% vs. 8%) [84]. Testing for dMMR is essential to identify this subset of older patients with CRC, as positive results provide prognostic information and are potentially predictive of benefit from immunotherapy.

Generally, dMMR is associated with lower recurrence risks in stage II disease CRC, and adjuvant therapy is not recommended for these patients.

For stage III disease in the adjuvant setting, single-agent 5-FU has generally not been shown to be beneficial for patients with dMMR CRCl [85,86]. Two recent meta-analyses have, however, suggested an overall survival benefit from the addition of adjuvant chemotherapy in stage III dMMR colon cancer [87,88]. The addition of oxaliplatin to treatment in this overall population of dMMR patients appears beneficial. As previously discussed, the use of adjuvant oxaliplatin remains debated in elderly patients. In an ACCENT pooled analysis, only 37 patients over the age of 70 with dMMR colon cancer were found to be randomly assigned to fluoropyrimidine versus fluoropyrimidine-oxaliplatin combination adjuvant therapy. Although these small numbers prevent addressing this important question for older adults, they did not identify a significant interaction between age and adjuvant oxaliplatin benefit in patients with dMMR disease [87]. Trials are ongoing to assess the role of immunotherapy in the adjuvant setting for these patients [89,90].

Rectal cancer will less commonly be dMMR, which is found in approximately 2.7% of cases [91]. The rates may be lower for older adults [92]. In June 2023, Cercek et al. published a phase II single-arm study of patients with dMMR stage II/III rectal cancer treated with dostarlimab (a PD-1 inhibitor) for 6 months. Remarkably, at initial publication, 12 of 12 patients had a clinically complete response to immunotherapy alone [93]. At the latest update, with 23 patients studied, a 100% clinical response rate still held. The oldest patient enrolled was 78 years of age. These results will require further follow-up, but the potential that the morbidity of cytotoxic chemotherapy, radiation, and surgery may be avoided in this select group of patients is very encouraging for both younger and older adults. This treatment strategy is reflected in recent guidelines and should not be restricted based on age [13]. 

Results from the use of immunotherapy in localized colon cancer with dMMR seem similarly impressive [94]. In the NICHE2 study, 112 patients (age 20–82) with cT3-4 and/or node positive colon cancer were treated with 2 cycles of nivolumab and a single cycle of ipilimumab prior to surgery. A pathological complete response rate of 67% and a major pathological response rate (<10% residual tumor volume) of 95% were reported. There is currently insufficient data to recommend neoadjuvant immunotherapy or surgery avoidance for the majority of patients with dMMR colon cancer. However, for advanced local diseases, or cases where surgery may be associated with higher risks of morbidity (as can be seen more frequently in less fit older adults), consideration of upfront immunotherapy may be reasonable [13,95].

## 6. Advanced/Metastatic Disease in Older Adults

22% of patients with CRC present with distant metastases [96], and many more will develop metastases during the course of their disease. Pathological evaluation of microsatellite status and next generation sequencing for RAS, BRAF, and HER2 gives important information for both prognostication and treatment decisions.

Colorectal cancer in the setting of limited metastases can often be treated with more aggressive approaches to potentially result in long-term survival or cure. The status of a patient’s metastatic disease falls into 3 categories determined by a multidisciplinary team of surgeons, radiologists, medical oncologists, and radiation oncologists: resectable, borderline resectable, and unresectable/palliative. As with all patients, establishing treatment goals is essential to making management decisions with older adults. Shared decision-making and CGA are key. It should be noted that older patients may hold quality of life as the highest priority [97], and the majority may not choose treatment if it would result in severe functional limitations [98].

Resectable disease involves situations where both the primary lesion and/or metastatic disease are amenable to curative resection. The 5-year survival rates following resection are generally between 20–45% for liver metastases and 25–35% for lung metastases [14]. Peritoneal resections are also possible in selected patients, leading to some long-term survivors [99,100]. Upfront surgery is a reasonable option for those with favorable disease features such as metachronous lesions, fewer metastases, and only unilobar liver disease [14]. Neoadjuvant chemotherapy for 3 months prior to surgical resection may offer a test of time and biology in those with less favorable disease characteristics. Generally, this involves fluoropyrimidine and oxaliplatin doublet chemotherapy. Due to the curative intent of this approach, a higher toxicity risk may be appropriate. It should be noted that despite the high risk of recurrence, neither perioperative FOLFOX [101,102] nor adjuvant FOLFOX [103] has been shown to increase overall survival in the setting of liver resection in randomized trials. 

Borderline resectable in this context means that the metastatic disease and primary tumor (if still present) are potentially resectable in the setting of an adequate response to systemic therapy. This is sometimes referred to as conversion therapy [14]. The selection of agents in this setting has generally been based on regimens associated with the highest response rates. However, the recently published CAIRO-5 trial in patients with unresectable liver metastases gives some randomized data in this setting [104]. For left-sided RAS/BRAF wild-type colorectal cancers, they found no difference in the rate of liver resection or progression-free survival when combining doublet chemotherapy with an anti-epidermal growth factor receptor (EGFR) antibody or an anti-vascular endothelial growth factor antibody (VEGF). This is despite the marked difference in response rate: 80% for panitumumab versus 53% for bevacizumab combinations. There was higher toxicity with the anti-EGFR antibody to consider. For right-sided or RAS/BRAF mutated disease, FOLFOXIRI and bevacizumab led to a higher rate of liver resection and higher PFS than 5-FU and irinotecan (FOLFIRI) and bevacizumab/FOLFOX and bevacizumab (10.6 months versus 9.0 months, HR 0.76, *p* = 0.032). In the TRIBE2 trial described below, similarly, 17% of patients treated with FOLFOXIRI and bevacizumab underwent resection of metastases with a negative margin versus 12% treated with FOLFOX and bevacizumab (*p* = 0.047) [105]. The decision to be aggressive with an approach to borderline resectable disease in older adults must weigh factors including the overall health of the patient, functional status, and social support that they may have. Unfortunately, there is a lack of specific data on older adults treated in this setting.

Palliative treatment aims to help the older patient with metastatic CRC that is not amenable to resection achieve the best quality of life possible. Systemic therapy has significant potential benefits in this setting. With supportive care alone, the median survival from metastatic CRC is around 5 months [106]. In common circumstances, this may improve more than 7-fold with systemic therapy in fit older patients [107]. Like in other settings, the patient’s fitness/frailty levels greatly influence treatment decisions and expected benefits.

Most commonly, treatment in the palliative setting involves combination chemotherapy (e.g. FOLFOX or FOLFIRI) combined with a biologic agent. 

## 7. Traditional Chemotherapy Backbones in a Palliative Setting

### 7.1. FOLFOX

In the pre-biologics era, the addition of oxaliplatin to 5-FU has shown a clear benefit in terms of overall survival in those >70 years old in pooled analyses of patients fit for clinical trials [108,109]. In elderly specific prospective phase II trials, this combination has similarly been shown to be safe and effective [110,111]. 

In vulnerable or frail older adults, there are a series of prospective elderly specific trials and analyses showing that treatment intensification is not beneficial for these patients. The FOCUS-2 study specifically enrolled patients with unresectable or metastatic CRC deemed “unfit” for full-dose chemotherapy by their medical oncologist [61]. Patients were randomized to receive capecitabine or 5-FU with or without oxaliplatin at a decreased dose (20% reduction). The addition of oxaliplatin did not lead to a statistically significant improvement in progression-free survival (PFS) or OS in this population. Oxaliplatin use was associated with a detrimental effect on quality of life. In the single-agent chemotherapy arms, capecitabine and 5-FU had similar efficacy; however, as mentioned above, capecitabine led to increased toxicity without improvement in quality of life compared to 5-FU.

### 7.2. FOLFIRI

Data suggest that in general, treating with FOLFIRI chemotherapy followed by FOLFOX at progression has similar outcomes to the reverse sequence [112]. A 2008 pooled analysis of 4 phase III trials comparing fluoropyrimidine alone to its combination with irinotecan in the first-line setting showed no relationship between age and treatment effect [112]. There was a trend towards improved overall survival with the addition of irinotecan, but this did not reach significance. Regression analysis suggested more diarrhea side-effects with increasing age.

The FFCD 2001-02 trial was a randomized phase III study that evaluated irinotecan combined with 5-FU versus 5-FU alone in patients with metastatic CRC age 75 years or older. PFS and OS were not improved by the addition of irinotecan in this trial, nor for any subgroup analyzed [113]. Grade 3–4 toxicity was higher with the addition of irinotecan (76.3%) than 5-FU alone (52.2%). They incorporated assessment of geriatric factors for a subset of patients and found that in addition to irinotecan use, a mini-mental state examination score of <27/30 or a baseline impairment in IADLs had an OR of 5.43 (range, 2.09 to 14.11; *p* < 0.001) for grade 3–4 toxicity [114]. Baseline independence of IADLs was predictive of better OS but did not benefit from irinotecan [115]. Thus, similar to combination therapy with FOLFOX, less fit older patients may not be appropriate candidates for combination irinotecan plus 5-FU. In fit older adults, there may be benefits from FOLFIRI, but there remains some uncertainty.

### 7.3. Anti-VEGF Antibody

Pooled analyses of randomized trials have found evidence that the addition of the anti-vascular endothelial growth factor antibody bevacizumab leads to increased survival in older patients [116,117]. The AVEX trial was a randomized trial of elderly patients ≥ 70 years of age deemed not to be candidates for doublet chemotherapy [118]. The addition of bevacizumab to single-agent capecitabine met the primary endpoint of PFS (9.1 versus 5.1 months, HR 0.53, *p* < 0.001). The trial was not sufficiently powered for OS assessment (20.7 versus 16.8 months, HR 0.79, *p* = 0.182). The safety and efficacy of chemotherapy plus bevacizumab were also demonstrated in the phase II PRODIGE 20 trial, which specifically enrolled patients aged 75 and older [119]. A review concluded that bevacizumab has a similar safety profile in younger and older adults, with the exception of a slight increase in arterial thrombotic risk with age [120].

The phase III JCOG1018 RESPECT trial randomized elderly patients with unresectable metastatic CRC to FOLFOX/CAPOX plus bevacizumab versus capecitabine or 5-FU plus bevacizumab [121]. A total of 93% of patients were >75 years of age with a performance status of 0–1. In the preliminary results, they found no difference in median PFS or OS from the addition of oxaliplatin to fluoropyrimidine and bevacizumab. The rates of grade 2–4 neutropenia, nausea, diarrhea, fatigue, and neuropathy were higher with oxaliplatin. Certainly, these results seem to further support omitting oxaliplatin in more frail older adults and may lower the threshold to drop oxaliplatin at the onset of toxicity in fit older adults receiving 5-FU and bevacizumab. These preliminary results are not sufficient to warrant the omission of oxaliplatin for all older patients in the front-line setting in light of previous data.

Trifluridine–tipiracil in combination with bevacizumab was not superior to capecitabine-bevacizumab for patients who were poor candidates for doublet chemotherapy. PFS was similar, with 9.4 months for trifluridine–tipiracil plus bevacizumab versus 9.3 months for capecitabine plus bevacizumab [122]. Post hoc analysis in patients 70 years of age and older did suggest favorable results with trifluridine–tipiracil plus bevacizumab in this population. This combination may be reasonably considered in older adults with contraindications to fluorouracil or capecitabine [122,123].

### 7.4. Anti-EGFR Antibody

Anti-EGFR therapy is associated with an improvement in progression-free and overall survival in metastatic colorectal cancer in those with left-sided primary tumors without RAS, BRAF, or HER2 alterations [107,124]. A recent pooled analysis of 7 randomized trials compared doublet chemotherapy (117 patients) to doublet chemotherapy with the addition of anti-EGFR therapy (123 patients) in those >70 years of age [125]. Older patients had inferior PFS and OS compared to younger patients. The analysis showed a non-statistically significant improvement in OS (24.7 vs. 17.6 months; HR 0.77, *p* = 0.092) and PFS (9.1 vs. 8.7 months; HR 0.85, *p* = 0.287) from the addition of anti-EGFR therapy to the treatment of adults > 70, after adjusting for key confounders. They, however, found no evidence that the treatment effect was based on age. Older patients did not have statistically higher rates of grade 3+ neutropenia, diarrhea, nausea/vomiting, or neuropathy, suggesting doublet chemotherapy and anti-EGFR therapy can be safely administered.

The phase II PANDA study enrolled patients > 70 years of age with unresectable metastatic RAS-BRAF wild-type colorectal cancer [10]. All patients underwent initial G8 screening, and this was a stratification factor for randomization. A total of 185 patients were randomized to 5-FU and panitumumab with or without oxaliplatin for 12 cycles, followed by panitumumab maintenance. Although this trial was not designed to directly compare between the two arms, there was a notable reduction in rates of grade 3–4 toxicities: FOLFOX–panitumumab/5-FU-panitumumab: neutropenia 10%/1%; diarrhea 16%/1%; stomatitis 10%/4%; neurotoxicity 3%/0%; fatigue 8%/4%. Grade 3–4 hypomagnesemia was surprisingly higher in the arm without oxaliplatin (3%/8%), and skin toxicities were similar (25%/24%) between the two groups. PFS appeared similar in both groups: 9.6 months for FOLFOX–panitumumab and 9.0 months for 5-FU-panitumumab. Median OS was 23.5 months in the FOLFOX–panitumumab arm and 22.0 months in the 5-FU-panitumumab arm (HR 1.0, *p* = 0.986). There was a significant interaction between age subgroups (70–75 versus >75 years) and OS, with better outcomes in younger patients treated with FOLFOX–panitumumab versus 5-FU-panitumumab. Similar to the results from the RESPECT trial, these findings may give some comfort that oxaliplatin can be omitted to reduce toxicity without major compromise to efficacy in older patients treated with biologic therapy in combination with fluoropyrimidine.

### 7.5. Triplet Chemotherapy and Biologic

The use of triplet (5-FU, irinotecan, and oxaliplatin) chemotherapy and bevacizumab in the first-line metastatic setting is popular in certain countries. The FOLFOXIRI plus bevacizumab was compared to the FOLFIRI plus bevacizumab in the original phase III TRIBE study [126]. They did show a benefit in OS (29.8 months with FOLFOXIRI plus bevacizumab vs. 25.8 months in the control, HR 0.8, *p* = 0.03), which was a secondary endpoint of the study [105]. The TRIBE2 study is a similar comparison between FOLFOXIRI plus bevacizumab versus FOLFOX plus bevacizumab [127]. Patients were treated for 8 cycles, followed by 5-FU and bevacizumab maintenance. At first progression, the experimental arm was rechallenged with FOLFOXIRI and bevacizumab, whereas the control arm was transitioned to FOLFIRI and bevacizumab at progression. The study showed an improvement in the PFS2 primary endpoint (defined as time until progression on any treatment given after first disease progression) of 19.2 months in the FOLFOXIRI plus bevacizumab arm versus 16.2 months in the control (HR 0.74, *p* = 0.0005). Median overall survival also favored the FOLFOXIRI plus bevacizumab arm (27.4 vs. 22.5 months, HR 0.82, *p* = 0.032). The TRIBE studies included patients up to the age of 70 that had ECOG 0-2, but only those with ECOG 0 were eligible between the ages of 71–75. A pooled analysis of the two trials showed no difference in response rates or PFS between patients older and younger than 70 years of age and no interaction between treatment and age [128]. However, older patients had higher rates of grade ≥ 3 adverse events (73% versus 60% in those <70 years of age, *p* < 0.01), including 27% diarrhea and 16% febrile neutropenia. These increased toxicities likely should lead to caution when using this regimen in older adults, and it should be reserved for the fittest patients under 75 years of age [14]. 

The combination of anti-EGFR therapy with triplet chemotherapy is not recommended in guidelines [14,129,130].

### 7.6. Immunotherapy

Pembrolizumab has become the standard first-line therapy for dMMR metastatic CRC based on the results of the Keynote 177 trial [131]. PFS was doubled when compared to first-line chemotherapy (16.5 months versus 8.2 months, HR 0.59). Due to a 60% rate of crossover, no overall survival benefit was established at the final analysis [132]. However, given that treatment-related adverse events were three times higher in the chemotherapy group, immunotherapy is clearly justified in the frontline setting for the majority of these patients. 

There were 90 patients over 70 years of age included in the trial, with the forest plot for PFS suggesting some attenuation of the benefit from pembrolizumab over chemotherapy in this age group (HR 0.52 for <70 years of age; HR 0.77 for those >70 years of age). There were similar findings for OS results. However, a growing body of literature has suggested that advanced age does not preclude and may even increase the benefit of immunotherapy in a variety of cancers [133]. In a recently published retrospective cohort study of mostly older adults with sporadic dMMR metastatic CRC (n = 41 patients, median age 81, 22% with ECOG 2-3) treated with first-line pembrolizumab, the response rate was 49%, the median PFS was 21 months, and the median OS was 36 months [134]. These results are on par with those of the Keynote 177 population as a whole. Immunotherapy should be considered in the front-line setting for metastatic disease in older patients with dMMR colorectal cancer. 

Combination treatment with nivolumab and ipilimumab shows promising results in metastatic dMMR CRC from the phase III CheckMate 8HW study with an HR of 0.21 compared to standard chemotherapy [135]. Although data from other cancers suggest combination immunotherapy treatment has a similar safety profile in older and younger patients [136], we have yet to clearly establish the superiority of combination therapy compared with single-agent immunotherapy in metastatic dMMR CRC.

### 7.7. Later-Line Options

More therapeutic options beyond fluoropyrimidine-based chemotherapy combinations (and pembrolizumab for patients with dMMR CRC) have been added to the colorectal cancer landscape in recent years.

The BEACON trial established encorafenib plus cetuximab as a preferred second- or third-line option for BRAF-mutated CRC due to improvement in overall survival and time until decline in quality of life [137,138]. The benefit was present in the subgroup of patients 65 years of age and older [139]. Dose interruptions did not differ significantly between those above and below 70 years of age, although nausea/vomiting, abdominal pain, and fatigue/asthenia were more common in older patients [140]. 

The RECOURSE trial showed a modest benefit in overall survival for trifluridine–tipiracil over placebo after exposure to standard chemotherapy and antibody therapies [141]. Subgroup analysis suggested the benefits were similar for patients aged ≥65 and ≥70 years [142]. The recently published SUNLIGHT trial showed the additional benefit of adding bevacizumab to trifluridine–tipiracil [143]. The primary endpoint of the trial was OS, with the median being 10.8 months for the trifluridine–tipiracil plus bevacizumab group compared to 7.5 months with trifluridine–tipiracil alone. 44% of participants were 65 years of age or older, and survival benefits were present irrespective of age.

The CORRECT trial showed a 1.4-month survival benefit for regorafenib (160 mg daily for three weeks on, one week off) in a later-line setting over best supportive care [144]. This small benefit seemed to be independent of age [145]. Alternative dosing schedules have been explored to reduce toxicity [146]. A small prospective trial of 23 patients over 75 years of age who were not frail by G8 screening/CGA suggested reasonable safety and efficacy when using a dose-escalation strategy and scheduling of 2 weeks on and 1 week off of regorafenib [147]. 

A new oral tyrosine kinase inhibitor of vascular endothelial growth factor receptors, fruquintinib, has shown efficacy in patients exposed to or intolerant of the above-mentioned therapies, where appropriate. In the FRESCO-2 trial, there was a 2.6-month OS benefit compared to the placebo established (7.4 months for fruquintinib versus 4.8 months for placebo) [148]. The benefit appeared to be present in patients older than 65 years of age. Overall, this appears to be a well-tolerated medication, although there were some increased rates of hypertension, asthenia, and hand-and-foot syndrome compared to placebo. Food and Drug Administration approval was granted on 8 November 2023.

## 8. Conclusions

A summary of treatment recommendations is provided in Table 4. Where available, these recommendations are derived from elderly specific prospective data (Table 5). Unfortunately, many recommendations are ultimately derived from trials of mostly younger populations that have rapidly changed the standard of care for patients with CRC.

Chronological age alone is not a reason to withhold potentially effective therapy. Geriatric assessment is more useful in helping with treatment decisions and limiting toxicity for older individuals with CRC. CGA should become routinely used in this setting. For those who are frail with severe, irreversible limitations (unfit), it is appropriate to opt against cancer-directed therapies. CGA results and clinical scenarios will inform decisions to escalate or de-escalate therapy for older adults with colorectal cancer who are candidates for treatment.

## 9. Future Directions

In the future, the expansion of minimally invasive techniques, prognostic tools such as ctDNA, and refinement of immunotherapy use in dMMR disease may further reduce the morbidity associated with CRC management. A significant portion of CRC cases involve patients of advanced age, and they warrant specific attention and inclusion in the design of future clinical trials. Further prospective validation of geriatric assessment tools to guide treatment selection, specifically in the context of CRC, is needed. Obtaining robust quality of life and toxicity data in future studies is essential. This is especially true given the priorities of older adults with CRC: to better inform shared decision-making.

## Figures and Tables

**Table 1 cancers-16-01820-t001:** Geriatric Screening Tools.

Tool	Purpose	Components	Scoring	CRC-Specific Evidence
G8 [8]	To identify patients in need of CGA	Decline in food intake; weight loss; mobility impairment; neuropsychological problems; BMI; polypharmacy; patient perceived health status; age	G8 > 14—CGA not necessary G8 ≤ 14—CGA necessary	- In PANDA study, G8 scores were prognostic for survival; however, there was no interaction between treatment arm (irinotecan or no irinotecan) and G8 score for PFS, ORR, OS, or grade > 2 toxicity [10] - In patients ≥ 75, lower G8 score is associated with higher rates of chemotherapy dismissal, dose reduction, and poorer overall survival [11,12]
CRASH [13]	To predict hematological and non-hematological treatment-related toxicity	Diastolic BP, IADLs, LDH, MMHS, chemotherapy risk	Hematological: 0–1—low; 2–3—low-intermediate; 4–5-intermediate-high; ≥6—high Non-hematological: 0–2—low; 3–4—low-intermediate; 5–6—intermediate-high; ≥7—high Combined score 0–3—low; 4–6—low-intermediate; 7–9—intermediate-high; ≥10—high	- In PANDA study, CRASH scores were prognostic for survival; however, there was no interaction between treatment arm (irinotecan or no irinotecan) and CRASH score for PFS, ORR, OS, or grade > 2 toxicity [10]

CGA—comprehensive geriatric assessment; BMI—body mass index; ECOG—Eastern Cooperative Oncology Group performance; MMHS—mini-mental health status; BP—blood pressure; LDH—lactate dehydrogenase; IADLs—instrumental activities of daily living; PFS—progression-free survival; ORR—objective response rate; OS—overall survival.

**Table 2 cancers-16-01820-t002:** Components of the Comprehensive Geriatric Assessment.

Domain	Deficit	Interventions
Functional Status	Limitations in basic activities of daily living or instrumental activities of daily living Timed Up and GO > 13 s [13] Falls history	Home safety evaluation Physiotherapy Occupational therapy Gait strengthening
Comorbidities	Comorbid conditions Hearing and visual impairments Pre-existing neuropathy	Co-management with primary care provider Referrals to subspecialty services
Cognition	Memory loss/impairment Confusion	Formal cognitive testing [16,17] Delirium prevention Capacity assessment Involvement of caregivers
Nutrition	Weight loss > 5% Iron deficiency or B12 deficiency anemia Problems with eating	Mini-Nutritional Assessment [18] Dietician involvement Supplementation Medications (mirtazapine or olanzapine are helpful in some circumstances) Speech and language pathologist for swallowing assessment
Psychological Status	Feeling sad or depressed Anxiety	Geriatric Depression Scale [19,20] Psychiatry referral Counseling Chaplaincy referral
Social Circumstances	Patient lives alone Lack of social support Barriers to social activity	Social work referral Community resources Meal/transportation programs
Polypharmacy	≥5 Prescribed medications ≥1 Supplement	Pharmacy review of medications for interactions Discontinuation of unnecessary medications
Clinical Symptoms	Pain Nausea Incontinence Diarrhea or constipation Neuropathy	Supportive care/pain management referral Single prescriber for opioids Educational interventions

Adapted from Li et al. [21]—Geriatric Assessment-Driven Intervention (GAIN) study.

**Table 3 cancers-16-01820-t003:** Examples of randomized trials of comprehensive geriatric assessment-driven interventions.

Study	Design (n)	Inclusion Criteria	Intervention	Efficacy	Comments
GAIN [21]	2:1 Randomized Trial (605)	Age ≥ 65, solid malignancy, starting chemotherapy	A: CGA-guided management and interventions (see Table 2 for examples) B: Standard care (no CGA-guided recommendations)	1°: Grade ≥ 3 chemotherapy-related toxicity: 50.5% vs. 60.6%, *p* = 0.02Advanced Directive completion: 28.4% vs. 13.3%, *p* < 0.00112 m OS: 66% vs. 64%; *p* = 0.55	- 33% of patients had GI cancers - reduction in toxicities occurred despite no differences in dose intensity, suggesting benefit from other geriatric-specific interventions
GAP70+ [28]	Cluster-randomized trial (718)	Age ≥ 70, incurable solid malignancy or lymphoma, ≥1 geriatric domain impairment (other than poly- pharmacy), starting a new systemic regimen with ≥50% prevalence of grade 3–5 toxicity	A: CGA summary and management recommendations B: Standard care (no recommendations)	1°: Grade ≥ 3 toxicity: 51% vs. 71%; *p* < 0.001Patient-reported symptomatic toxicities Grade ≥ 2: 88.9% vs. 94.8%; *p* = 0.0356-month OS: 72% vs. 75%; *p* = 0.38	- first cycle dose reductions are more common in the CGA group - fewer falls in the CGA group and more medication discontinuation
INTEGERATE [29]	1:1 randomized trial (154)	Age ≥ 70, solid malignancy or diffuse-large B-cell lymphoma, starting systemic therapy	A: CGA B: Standard care	1°: Change in health-related quality of life: better in the intervention group, *p* = 0.030 - Reduction in unplanned hospital admissions in the intervention group (*p* = 0.007)	- CGA was performed by a single dual trained geriatrician-oncologist, which may limit external validity

1°—primary endpoint of study; CGA—comprehensive geriatric assessment; GI—gastrointestinal; OS—overall survival; m—month.

**Table 4 cancers-16-01820-t004:** Summary of recommendations for treating older adults with colorectal cancer.

Clinical Scenario				Recommendations
**All Stages of Disease**				
				Geriatric screening tools such as the G8 should be used to identify patients who will benefit from a comprehensive geriatric assessment A comprehensive geriatric assessment should be performed when possible in all older adults who score < 14 on the G8 screening or have other clear health, functional, or social limitations Multidisciplinary discussion between oncology (surgeon, radiation oncologist, medical oncologist) and geriatric specialists Enrollment in clinical trials whenever possible Unfit patients with limited life expectancy or other life-limiting conditions should be managed with the best supportive interventions
**Localized Disease, Microsatellite Stable**				
	Early Stage Rectal Cancer (T1-2, N0)			
			FIT	Transanal excision for feasible T1 tumors
			MEDIUM FIT	Transanal excision for feasible T1 tumors Short-course radiation followed by transanal endoscopic microsurgery is a reasonable initial approach for T2 tumors
	Locally Advanced Rectal Cancer (T3-4, N0-2, or TxN+), Neoadjuvant Therapy			
			FIT	TNT approaches are generally favored over adjuvant treatments. cT3 mid-high tumors or cT2 tumors without compromised mesorectal fascia, however, may be candidates for upfront surgery cT2N1, or T3N0-1 without compromised mesorectal fascia—omission of radiation prior to surgery in those responding to FOLFOX does not compromise outcomes Sequential FOLFIRINOX, CRT, surgery, and adjuvant chemotherapy can be considered only for very fit patients < 75 years of age Short-course radiation followed by FOLFOX prior to surgery can be considered for those with high-risk features Discuss watch-and-wait approaches for those with low tumors and a complete clinical response to neoadjuvant therapy. If the goal is to avoid surgery, success rates are higher when CRT is given before chemotherapy
			MEDIUM FIT	CRT followed by transanal endoscopic microsurgery may be a reasonable initial approach for T3N0 tumors with ECOG 2. There are high risks of lower anterior resection syndrome with this approach TNT approaches are generally favored over adjuvant treatments. cT3 mid-high tumors or cT2 tumors without compromised mesorectal fascia, however, may be candidates for upfront surgery Short-course radiation followed by total mesorectal excision may have short-term quality of life benefits over CRT followed by TME Discuss watch and wait approaches for those with low tumors and a complete clinical response to neoadjuvant therapy
	Colon Cancer, Adjuvant Therapy			
		Stage II with high-risk features		
			FIT or MEDIUM FIT	Consider 6 months of single-agent capecitabine/5-FU Omission of chemotherapy may be reasonable. Negative ct-DNA may further support the decision to omit chemotherapy, particularly in those without T4 disease
		Stage III		
			FIT	Low risk: 6 months of capecitabine/5-FU or 3 months of CAPOX or 6 months of capecitabine/5-FU High risk (T4 or N2): Consider 3–6 months of CAPOX or 6 months of FOLFOX. The threshold to drop oxaliplatin beyond 3 months may be low Shared decision-making as a benefit of adjuvant oxaliplatin is unclear in older adults
			MEDIUM FIT	6 months of capecitabine/5-FU
**Localized Disease, Microsatellite Instability**				
	Rectal Cancer, Neoadjuvant Setting			
			FIT or MEDIUM FIT	Upfront immunotherapy with consideration of surgical avoidance in the setting of a complete response
	Colon Cancer, Adjuvant Setting			
		Stage II		
			FIT or MEDIUM FIT	Adjuvant chemotherapy is generally not recommended due to the good prognosis and lack of benefit
		Stage III		
			FIT	Consider CAPOX or FOLFOX for 3–6 months
			MEDIUM FIT	The benefit of adjuvant chemotherapy is unclear in those not fit for oxaliplatin, given the potential harm from single-agent capecitabine/5-FU
**Metastatic Disease**				
		Resectable		
			FIT	6 months of perioperative FOLFOX, or proceed directly to resection Perioperative/adjuvant chemotherapy has not shown survival benefit
			MEDIUM FIT	Proceed directly to surgery, or consideration of single-agent 5-FU or capecitabine Perioperative/adjuvant chemotherapy has not shown survival benefit
		Borderline Resectable		
			FIT	Consider FOLFOXIRI plus bevacizumab for very fit patients < 75 years of age Doublet chemotherapy plus biologic is reasonable
			MEDIUM FIT	Doublet chemotherapy and plus biologic with initial dose reduction, or single-agent 5-FU with biologic
		Palliative Setting, First-Line		
			FIT	Left-sided primary, RAS/BRAF/HER2 wt: doublet chemotherapy with anti-EGFR therapy. Discussion about the unclear benefit of oxaliplatin/irinotecan with anti-EGFR-fluoropyrimidine in older patients Right-sided primary, or RAS/BRAF/HER mutant: doublet chemotherapy with bevacizumab. Discussion about the unclear benefit of oxaliplatin/irinotecan with bevacizumab-fluoropyrimidine in older patients FOLFOXIRI with bevacizumab can only be considered for the fittest patients < 75 years of age after discussion of increased toxicity risk Pembrolizumab for dMMR disease
			MEDIUM FIT	Left-sided primary, RAS/BRAF/HER2 wt: 5-FU/capecitabine with anti-EGFR therapy Right-sided primary, or RAS/BRAF/HER mutant: 5-FU/capecitabine +/− bevacizumab Trifluridine/tipiracil + bevacizumab in setting of 5-FU/capecitabine contraindication Pembrolizumab for dMMR disease
		Palliative Setting, Later Lines		
			FIT or MEDIUM FIT	BRAF mutated: encorafenib-cetuximab is a second- or third-line option Trifluridine/tipiracil with bevacizumab is an effective option Regorafenib may be appropriate for fit patients, but only after weighing the minimal survival benefit with the toxicity profile. Alternative dosing strategies should be employed Fruqintinib is an appropriate option, preferably after trifluridine/tipiracil or regorafenib

**Table 5 cancers-16-01820-t005:** Prospective randomized trials focused on elderly or frail populations with colorectal cancer.

Study	Phase (n)	Inclusion Criteria (Median Age)	Intervention	Efficacy	Comments
RESPECT (JCOG1018) [121]	III (251)	mCRC, Age 70–74 with PS2 or ≥75 with PS 0-2, first-line (79)	A: FOLFOX/CAPOX plus bevacizumab B: 5-FU/Capecitabine plus bevacizumab	1°: OS: 21.3 vs. 19.7 m HR 1.054 (0.810–1.372) PFS: 9.4 vs. 9.3 m; *p* = 0.086 ORR: 47.7% vs. 29.5%	- Higher rates of adverse events and numerically fewer improvements in QoL with the addition of oxaliplatin
PANDA [10]	II (183)	mCRC, first-line, RAS/BRAF-wild type, age 70–75 with PS < 2, or age > 75 years with PS < 1 (77)	A: FOLFOX plus panitumumab B: 5-FU plus panitumumab	1°: Non-comparator trial, to reject null hypothesis of PFS < 6 m; *p* < 0.001 in each arm PFS: 9.6 vs. 9.1 m, *p* = 0.611 (unplanned comparison) OS: 23.5 vs. 22.0 m, *p* = 0.986 ORR: 69% vs. 52%; *p* = 0.182	- Higher rates of adverse events with the addition of oxaliplatin - 5FU plus panitumumab had poor PFS and is not a good choice for right-sided tumors
SOLSTICE [122]	III (856)	mCRC, first-line, not candidates for full-dose doublet or triplet chemotherapy (73)	A: TAS-102 plus bevacizumab B: Capecitabine plus bevacizumab	1°: PFS 9.4 vs. 9.3 m; *p* = 0.464 OS: immature ORR: 36% vs. 42%; *p* = 0.092	- Global health-related quality of life scores were similar - more cytopenias with TAS-102; more hand-and-foot syndrome with capecitabine
AVEX [118]	III (280)	mCRC, first-line, age ≥ 70, not candidates for doublet chemotherapy (76)	A: Capecitabine plus bevacizumab B: Capecitabine	1°: PFS: 9.1 vs. 5.1 m; *p* < 0.001 OS: 20.7 vs. 16.8 m; *p* = 0.18 ORR: 19% vs. 10%; *p* = 0.04	- Safety profile of bevacizumab is similar for older and younger adults
MRC FOCUS2 [61]	Randomized 2X2 factorial trial (459)	mCRC, first-line, not candidates for full-dose doublet chemotherapy (74)	A: FOLFOX (80% dose, with bolus) B: CAPOX (80% dose) C: 5-FU (80% dose) D: Capecitabine (80% dose)	1°: PFS: [A vs. C] + [B vs. D] 5.8 vs. 4.5 m; *p* = 0.07 OS: No evidence of benefit with oxaliplatin ORR: 41.2% vs. 12.1%	- Oxaliplatin was associated with a detrimental effect on quality of life in this frail population
FFCD 2001-02 [113]	III, Randomized 2X2 factorial (282)	mCRC, first-line, age ≥ 75 (80)	A,B: 5-FU + irinotecan arms C,D: 5-FU arms	1°: PFS: 7.3 vs. 5.2 m; *p* = 0.15 OS: 13.3 vs. 15.2 m; *p* = 0.77 ORR: 25.6% vs. 37.1%; *p* = 0.04	- impaired MMS, impairment in IADL, and irinotecan use are associated with higher grade 3–4 toxicities
PRODIGE 42/GERICO 12 [41]	III (103)	Locally advanced rectal cancer (very low cT2 or at least cT3), age ≥ 75 (80)	A: Long-course chemoradiation B: short-course radiation	1°: R0 resection rate: 95.6% vs. 88.0%; *p* = 0.49, non-inferiority not reached 1°: Deterioration of autonomy at 12 months: 28.6% vs. 29.2%; not statistically significant OS: HR 0.28; *p* = 0.05 favoring short-course radiation RFS: HR 0.99; *p* = 1 pCR: 17.4% vs. 6.1%; *p* = 0.16	- failed to meet enrollment target - underpowered for overall survival

mCRC—metastatic colorectal cancer; PS—performance status; 5-FU—5-fluoruracil; FOLFOX—5-FU and folinic acid with oxaliplatin; FOLFIRI—5-FU and folinic acid with irinotecan; CAPOX—capecitabine with oxaliplatin; OS—overall survival; PFS—progression-free survival; ORR—objective response rate; MMS—mini-mental state exam score; TAS-102—trifluridine/tipiracil; IADL—instrumental activities of daily living; RFS—recurrence-free survival; pCR—pathological complete response.

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
