# Peer review of "Updates on the Management of Colorectal Cancer in Older Adults"

_cancers, 2024, doi:10.3390/cancers16101820_

Round 1
Reviewer 1 Report
Comments and Suggestions for Authors
Congratulations on an excellent review of the subject. Perhaps another table should show the proportion and the number of patients in this age group in each trial mentioned and stress the weight of current evidence on the topic.
Author Response
Thank you very much for the kind comments and constructive feedback.
We have added a new table (Table 5) to include the most important prospective randomized trials that have specically included elderly/frail populations. This includes information about the quality of the evidence (trial phase, number of patients, age of patients and whether primary endpoint was met).
Reviewer 2 Report
Comments and Suggestions for Authors
This is a well-written comprehensive review that focuses on geriatric management of colorectal cancer. The authors assimilated prospective data from recent fourteen clinical trials that involved older adults. The strength lies in the fact that the project highlighted the management pattern of a particular age group in this cancer category. I would like to recommend including a table that shows different agents or management strategies that were used in these trials.
Author Response
Thank you very much for the comments and suggestions.
We have added 3 tables to further elaborate on the different "agents"/tools and management strategies used in geriatric-specific care:
Table 1 - contains details on geriatric screening tools.
Table 3 - contains some trials demonstrating the effects of compressive geriatric assessment in various cancers.
Table 5 - contains the prospective, elderly-specific, colorectal cancer randomized trials assessing different agents.
Reviewer 3 Report
Comments and Suggestions for Authors
This is a very well written, logically structured article.
The authors have examined treatment strategies for CRCs in older age in the light of recent findings, taking into account the biological age of the patient.
They also made recommendations relevant to practical patient care for both biologically fit and less fit patients, in such a way that patients benefit as much as possible from the treatment strategy proposed for them.
The professional background, the literature used and the interpretation of the studies considered are excellent.
Table 2 is particularly useful.
I have one small question: should all elderly patients necessarily be treated? When is best supportive care recommended?
Author Response
Thank you very much for the kind comments.
We have added a line to the conclusion section to further stress that for elderly patients that are unfit, cancer-directed treatment should be withheld.